# Post-Infectious Guillain–Barré Syndrome Related to SARS-CoV-2 Infection: A Systematic Review

**DOI:** 10.3390/life11020167

**Published:** 2021-02-21

**Authors:** Pasquale Sansone, Luca Gregorio Giaccari, Caterina Aurilio, Francesco Coppolino, Valentina Esposito, Marco Fiore, Antonella Paladini, Maria Beatrice Passavanti, Vincenzo Pota, Maria Caterina Pace

**Affiliations:** 1Department of Woman, Child and General and Specialized Surgery, University of Campania Luigi Vanvitelli, 80138 Napoli, Italy; lucagregorio.giaccari@gmail.com (L.G.G.); Caterina.aurilio@unicampania.it (C.A.); francesco.coppolino1987@gmail.com (F.C.); valentina.mge@gmail.com (V.E.); mariabeatrice.passavanti@unicampania.it (M.B.P.); vincenzo.pota@inwind.it (V.P.); caterina.pace@libero.it (M.C.P.); 2Cotugno Hospital, Azienda Ospedaliera dei Colli, 80131 Napoli, Italy; marco.fiore@unicampania.it; 3Department of MESVA, University of L’Aquila, 67100 L’Aquila, Italy; antonella.paladini@cc.univaq.it

**Keywords:** Guillain-Barré syndrome, Miller Fisher syndrome, severe acute respiratory syndrome coronavirus-2, SARS-CoV-2, COVID-19

## Abstract

*Background.* Guillain-Barré syndrome (GBS) is the most common cause of flaccid paralysis, with about 100,000 people developing the disorder every year worldwide. Recently, the incidence of GBS has increased during the severe acute respiratory syndrome coronavirus-2 (SARS-CoV-2) epidemics. We reviewed the literature to give a comprehensive overview of the demographic characteristics, clinical features, diagnostic investigations, and outcome of SARS-CoV-2-related GBS patients. *Methods.* Embase, MEDLINE, Google Scholar, and Cochrane Central Trials Register were systematically searched on 24 September 2020 for studies reporting on GBS secondary to COVID-19. *Results.* We identified 63 articles; we included 32 studies in our review. A total of 41 GBS cases with a confirmed or probable COVID-19 infection were reported: 26 of them were single case reports and 6 case series. Published studies on SARS-CoV-2-related GBS typically report a classic sensorimotor type of GBS often with a demyelinating electrophysiological subtype. Miller Fisher syndrome was reported in a quarter of the cases. In 78.1% of the cases, the response to immunomodulating therapy is favourable. The disease course is frequently severe and about one-third of the patients with SARS-CoV-2-associated GBS requires mechanical ventilation and Intensive Care Unit (ICU) admission. Rarely the outcome is poor or even fatal (10.8% of the cases). *Conclusion.* Clinical presentation, course, response to treatment, and outcome are similar in SARS-CoV-2-associated GBS and GBS due to other triggers.

## 1. Introduction

Guillain-Barré syndrome (GBS) is the most common cause of flaccid paralysis, with about 100,000 people developing the disorder every year worldwide [1]. GBS is considered a postinfectious disease as approximately two-thirds of patients report preceding infections with specific pathogens, such as Campylobacter jejuni (32%), cytomegalovirus (13%), and Epstein-Barr virus (10%) [2].

Recently, several cases of GBS were reported during the severe acute respiratory syndrome coronavirus-2 (SARS-CoV-2) epidemics worldwide. Patients with COVID-19 typically have fever and respiratory illness; however, a wide range of other symptoms have been described. While the neurological sequelae of the virus remain poorly understood, there are a growing number of reports of neurological manifestations of COVID-19 [3].

GBS is an acute, immune-mediated polyradiculoneuropathy with a wide range of clinical manifestations. The classic form of GBS is characterized by a rapidly progressive and symmetrical weakness of the limbs, with sensory symptoms and reduced or absent tendon reflexes [4]. The clinical presentation and severity of GBS can vary extensively between patients. Electrophysiological studies help confirm the diagnosis of GBS, and can indicate different subtypes, including acute inflammatory demyelinating polyradiculoneuropathy (AIDP), acute motor axonal neuropathy (AMAN), and acute motor and sensory axonal neuropathy (AMSAN) [5]. Besides the classic presentation of GBS, clinical variants are reported, such as the Miller Fisher syndrome (MFS) which is characterized by ophthalmoplegia, ataxia, and areflexia without any weakness [5].

Treatment with intravenous immunoglobulin (IVIg) or plasma exchange (PLEX) is the optimal management approach, alongside supportive care [6]. Most patients with GBS show extensive recovery, and about 80% of patients with GBS regain the ability to walk independently at 6 months after the disease onset [6].

Since the World Health Organization (WHO) declared COVID-19 as a “Public Health Emergency of International Concern” on 30 January 2020, the impact of COVID-19 on patients has been profound. As the full clinical spectrum of COVID-19 is continuing to be described, preliminary findings from case reports and case series have uncovered neurological complications. An important step is to get a better understanding of the acute and post-infectious manifestations of COVID-19 to guide long-term management and health service reorganization. To help add to this small albeit developing body of evidence, this systematic review adds to other studies on GBS secondary to COVID-19.

Objectives. We have performed a systematic review of all published studies on SARS-CoV-2-related GBS, and give a comprehensive overview of the demographic characteristics, clinical features, diagnostic investigations, and outcome of SARS-CoV-2-related GBS patients.

## 2. Materials and Methods

### 2.1. Protocol and Registration

We performed a systematic review based on the Preferred Reporting Items for Systematic Reviews and Meta-Analyses (PRISMA) statement [7]. The protocol was not published, and the review was not registered with the International prospective register of systematic reviews (PROSPERO).

### 2.2. Literature Search

We identified the articles by searching electronic databases (Embase, MEDLINE, Google Scholar, and Cochrane Central Trials Register). Other relevant studies were identified from the reference lists. We used a combination of such terms as “Guillain-Barré Syndrome” and “COVID-19”. The initial search was performed on 24 September 2020. The titles and abstracts were screened by two researchers (PS and LGG) to identify the key words. The selected papers were read in full by the two independent reviewers and a third reviewer (MCP) was consulted in case of disagreement.

We included all the papers with available full text, without any restriction of the year of publication, reporting original data of patients with GBS and a suspected, probable, or confirmed recent SARS-CoV-2 infection, of any age, gender, and in any setting.

Predefined exclusion criteria were: GBS within 3 months after a vaccination or other proven triggering infection (e.g., C. jejuni), and studies with no information on residence, and at least one clinical variable of interest.

### 2.3. Data Extraction and Management

Data were extracted independently by one of the three reviewers (PS, LGG, MCP) according to a predefined protocol. The data extraction was then checked by one of the other two reviewers, and discrepancies were solved by discussion among all of them. Variables of interest comprised demographics, clinical characteristics (symptoms and signs of coronavirus infection and GBS), ancillary diagnostic investigations (electrophysiology and cerebrospinal fluid [CSF]), treatment, clinical course, and outcome of GBS.

Cases were classified according to the reported diagnostic certainty levels for GBS and SARS-CoV-2 infection. To classify the diagnosis GBS, we employed the “Brighton Collaboration Criteria (2011)” [8]. These were defined on the basis of the available reported data. The diagnostic certainty of SARS-CoV-2 infection was established according to the World Health Organization (WHO) criteria [9].

Clinical characteristics were reported together with the ratio of the number of patients in whom the variable was present (n) and the total number of reported cases (N): n/N (%). As for the symptoms, we assumed they were absent rather than missing if they were not cited in the manuscript, in order to account for the reporting bias, and therefore described as zero (n) out of the total number of reported cases (N).

Continuous variables (age, time between infectious and neurological symptoms, duration of progression and plateau phase of GBS, duration of hospital admission) were extracted as medians and or means, depending on how they were presented in the original article.

## 3. Results

### 3.1. Study Selection

We identified 63 articles in the researched databases, and 32 of them were included in our systematic review. The 32 selected studies reported on a total of 41 GBS cases with a confirmed or probable COVID-19 infection: 26 of them were single case reports and 6 case series. The flow diagram (see Figure 1) shows the results from the literature search and the study selection process.

### 3.2. Study Characteristics 

All the Studies Are Presented Alphabetically with a Brief Clinical Description Per Case (Table 1). 

All the cases were from COVID-19 epidemic or endemic regions (see Figure 2): 12 cases were from Italy (29.3%), 7 from U.S. (17.1%), 6 from Spain (14.6%), 4 from France (9.7%), 3 from Iran (7.3%) and U.K. (7.3%), 2 from Germany (4.9%), and one from China (2.4%), Morocco (2.4%), Switzerland (2.4%), and Turkey (2.4%).

Thirty-three cases were positive for nasopharyngeal swab test for SARS-CoV-2 by qualitative RT-PCR assay, four for oropharyngeal swab test for SARS-CoV-2 by qualitative RT-PCR assay, two for anti-SARS-CoV-2 IgA, and four for anti-SARS-CoV-2 IgG. In Manganotti P et al., the patient was reported to be COVID-19 positive with no further information provided. After discussion, we decided to consider it as a probable case of SARS-CoV-2 infection.

A chest CT scan was performed in fifteen cases showing multiple bilateral ground glass opacities, typical of COVID-19 pneumonia. Eleven studies reported a chest X-ray: five studies showed bilateral ground glass pneumonia.

The classic form of GBS was diagnosed in 33 cases. Twenty-one of the twenty-seven cases where the Brighton classification was reported, fulfilled level 1. The most frequent clinical phenotype was demyelinating polyradiculoneuropathy. In 8 cases, the variant MFS was diagnosed.

### 3.3. Patient Characteristics

#### 3.3.1. Demographics 

The median age of the study populations varied between 36 and 74 years, and only 1 pediatric patient was included in Paybast S et al. The majority of patients were male (62.8%) and the male:female ratio of all studies combined was 1.69. Toscano G et al. did not report the patients’ age.

#### 3.3.2. Certainty Levels of GBS Diagnosis and Infection 

Separate proportions of each Brighton level (1–4) were obtained in twenty-one studies (27 cases): 21 cases fulfilling level 1; 5 level 2; 0 level 3; and 1 level 4. MFS was reported in seven studies: two studies from Italy (1 patient in Assini A et al., and 1 patient in Manganotti P et al.), from Spain (1 patient in Fernandez-Dominguez J et al., and 2 patients in Gutierrez-Ortiz C et al.), and from U.S. (1 patient in Dinkin M et al., and 1 patient in Lantos JE et al.), and one study from U.K. (1 patient in Ray A).

SARS-CoV-2 infection was confirmed in 40 (97.6%) and probable in 1 (2.4%) of all cases.

#### 3.3.3. Clinical Characteristics 

All studies reported the presence of clinical symptoms of SARS-CoV-2 infection (see Figure 3). Two or more symptoms were present in 87.5% of cases. The most common symptoms were fever (63.4%, 26/41 cases), cough (51.2%, 21/41 cases), ageusia (22%, 9/41), myalgia (19.5%, 8/41), asthenia (17.1%, 7/41), anosmia (17.1%, 7/41), diarrhea (14.6%, 6/41), and headache (12.2%, 5/41). Others reported symptoms were chills, dyspnea, dizziness, and pruriginous rash.

The median time between the start of infectious symptoms and neurological symptoms ranged from 23 to 8 days in the studies reporting on this, except for Dinkin M et al., Oguz-Akarsu E et al., and Paybast S et al.

Among neurological findings, almost all studies reported on sensory symptoms, tendon reflexes, and facial palsy, whereas other symptoms were reported less frequently. The most frequent neurological findings were hypo/areflexia (*n* = 34), paraesthesia (*n* = 26), and limb paresis (*n* = 18). Other frequent symptoms were facial palsy and ataxia in about one third of cases. Other symptoms were reported less frequently, as shown in Table 2.

### 3.4. Diagnostic Investigations 

PCR, principally from nasal or throat swab, was the most frequently performed test for SARS-CoV-2 diagnosis. Anti-SARS-CoV-2 IgA was positive in two cases (Coen M et al., and Naddaf E et al.), and anti-SARS-CoV-2 IgG in three cases (Coen M et al., Naddaf E et al., and Riva N et al.). In the CSF, SARS-CoV-2 PCR was negative in all tested cases.

Only eleven studies tested all the cases for other infections that have been associated with GBS (C. jejuni, CMV, EBV, hepatitis E virus, mycoplasma pneumoniae) and all the tested cases (11/41; 26.8%) were negative for recent infection. None of the studies tested for all of these pathogens.

CSF was examined in thirty-one studies, and information on protein level and cell count was provided by all of these. Increased protein level and albuminocytological dissociation were present in the vast majority of cases. Only three studies reported normal CSF cell count: 1 case in Oguz-Akarsu E et al., 1 case in Riva N et al., and two cases in Toscano G et al.

Electrophysiological studies were done in 30 (73.2%) of the reported cases. The most frequent electrophysiological subtype was AIDP in 56.7%, followed by AMSAN in 16.7% and AMAN in 13.3% of cases. In three studies (Ottaviani D et al., Paybast S et al., and Rana S et al.) electrophysiological studies showed a mixed pattern of demyelination and axonal damage. The investigations are reported in Table 3.

### 3.5. Treatment and Disease Progression 

As shown in Table 3, all the studies provided information on the treatment, except for Marta-Enguita J et al. Almost all cases were treated with an immunomodulating therapy. In Caamaño DSJ et al., patients received oral prednisone. Only two cases (Dinkin M et al. and Gutierrez-Ortiz C et al.) did not require any treatment.

Thirty patients were treated with IVIg. The preferred therapeutic regimen was intravenous immunoglobulin at 0.4 g/kg/day for 5 days. Plasma exchange was performed four times (Granger A et al., Naddaf E et al., Oguz-Akarsu E et al., and Paybast S et al.). In two cases (Pfefferkorn T et al. and Toscano G et al), IVIg and PLEX were both administered.

Admission to ICU was required in a third of the cases, as well as the mechanical ventilation. Death was infrequent and only two studies (Alberti P et al. and Marta-Enguita J et al.) reported the patient exitus.

## 4. Discussion

It is now known that SARS-CoV-2 can often affect central and peripheral nervous system, apart from the bronchopulmonary system [42].

The current study reports various cases of GBS or its variants in patients with SARS-CoV2 infection. Our systematic review shows that published studies on SARS-CoV-2-related GBS typically report a classic sensorimotor type of GBS often with a facial palsy and a demyelinating electrophysiological subtype.

The time between onset of infectious and neurological symptoms and negative PCR in most patients suggests a post-infectious mechanism rather than a direct infectious one.

The mechanism involving the peripheral nervous system in SARS-CoV-2-infected patients and the development of GBS is unknown. Since most patients developed GBS on average 10 days after the first non-neurological symptoms of the SARS-CoV-2 infection, a causal relation is quite likely.

However, in none of the included patients, specific SARS-CoV2 RNA was found in the CSF. This suggests that GBS is not triggered by a direct viral attack against the nerve roots but rather by an immune-mediated mechanism, such as antibody precipitation on myelin sheaths or axons [43]. COVID-19 is associated with a cytokine storm and a dysregulated immune response [44]. A mimicry between the epitopes on the surface of the virus and the ones on neuronal membranes is possible, because they are simultaneously attacked by the immune response, just like in the GBS due to C. jejunii [45]. It is also possible that the virus directly invades motor and sensory neurons since the presence of the virus was reported in neural and capillary endothelial cells in frontal lobe tissue obtained at post-mortem examination from a patient infected with SARS-CoV-2 [46].

Published studies on SARS-CoV-2-related GBS typically report a classic sensorimotor type of GBS often with a demyelinating electrophysiological subtype. The distribution of subtypes varies between countries. In Europe and the United States, AIDP affects 60–80% of people with GBS, while AMAN affects only a 6–7%. In Asia and Central and South America, that proportion is nearly 30–65%. This is related to the exposure to various infections and to the genetic characteristics of population [47]. This study found that in Western countries, the demyelinating subtype affected most of the people with GBS, while AMAN/AMSAN affected only a small number. MFS was reported in a quarter of the cases. 

A male preponderance was noted as with non-SARS-CoV-2-associated GBS. Similarly, elderly patients are more frequently affected than the younger. 

We noticed how GBS can arise in patients who are totally asymptomatic or with moderate symptoms of COVID-19. Furthermore, the severity of GBS is not correlated with that of COVID-19. GBS-related neurological symptoms have often overlapped with COVID19-related symptoms, due to the short time between the SARS-CoV-2 infection and the onset of GBS (8–23 days). It is therefore evident that this overlap contributes to a poor prognosis. 

Increased protein level and albuminocytological dissociation were present in most cases. This pattern distinguishes GBS from other conditions, such as lymphoma and poliomyelitis, in which both the protein and the cell count are elevated. Furthermore, protein level in CSF was shown to correlate with disease activity, progression, and response to the treatment [48].

None of the patients with reported CSF analysis had SARS-CoV-2 in the CSF. This is in contradiction with various case reports, which detected SARS-CoV-2 in the CSF [49,50]. A possible explanation is the lack of disruption of the blood-brain barrier that would allow SARS-CoV-2 to cross the CSF space [51].

Another possible explanation is the low sensitivity, around 60%, of the currently available real-time reverse-transcription polymerase chain reaction (rRT-PCR) [52].

In almost all cases, the response to immunomodulating therapy was favorable. Intravenous immunoglobulins and plasmapheresis are the two main immunotherapy treatments for GBS [53,54]. A five-day course of daily IVIg (0.4 g/kg/day) was the most common regime adopted. Plasma exchange was performed four times. In two cases, IVIg and PLEX were both administered but a combination of the two is not significantly better than either alone [53]. Only a patient received corticosteroids, even if it is shown that they are not effective in speeding recovery and could potentially delay recovery [55].

The disease course was frequently severe and about one-third of the patients with SARS-CoV-2-associated GBS required mechanical ventilation and ICU admission. Rarely the outcome was poor or even fatal.

Limitations. Our study has several limitations. First, the studies included in this systematic review are variable in study design and setting, selection criteria, diagnostic ascertainment, and reporting of variables, which are potential sources of bias. Second, given the concurrent manifestations of COVID-19 and GBS, it is possible that some COVID-19 symptoms could be attributed to GBS and vice versa, since both diseases affect the respiratory system. Third, such cases without results of nerve conduction were not categorized into GBS subtypes, since it was not possible to confirm whether the symptoms were due to demyelination or axonal damage.

## 5. Conclusions

The SARS-CoV-2 infection can cause GBS. The underlying mechanism leading to development of this condition is still unclear. The CSF is free of SARS-CoV2 RNA, therefore excluding a viral attack directed against the nerve roots, but various hypotheses are possible. Clinical presentation, course, response to treatment, and outcome are similar in SARS-CoV-2-associated GBS and GBS due to other triggers. It is important to be aware of this association to avoid delay in diagnosis and to promote early treatment initiation given the significant risk of morbidity and mortality.

## Figures and Tables

**Figure 1 life-11-00167-f001:**
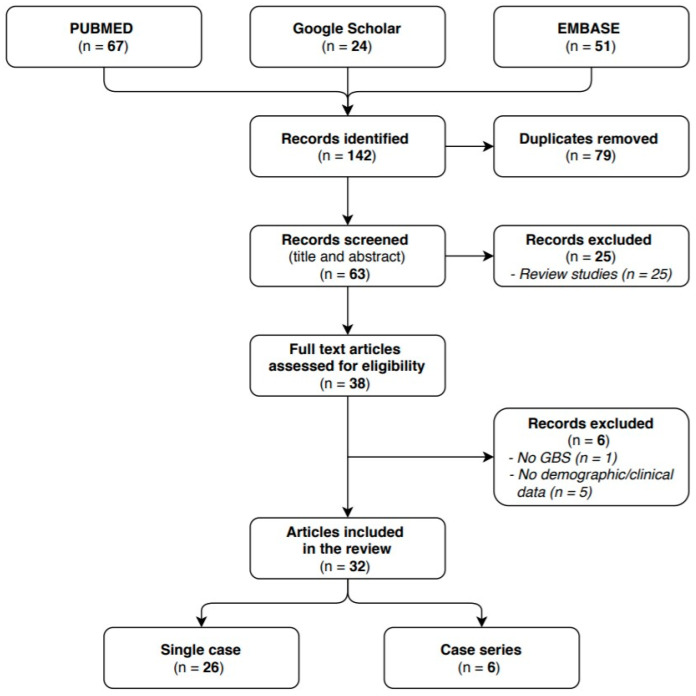
Flow diagram study selection process.

**Figure 2 life-11-00167-f002:**
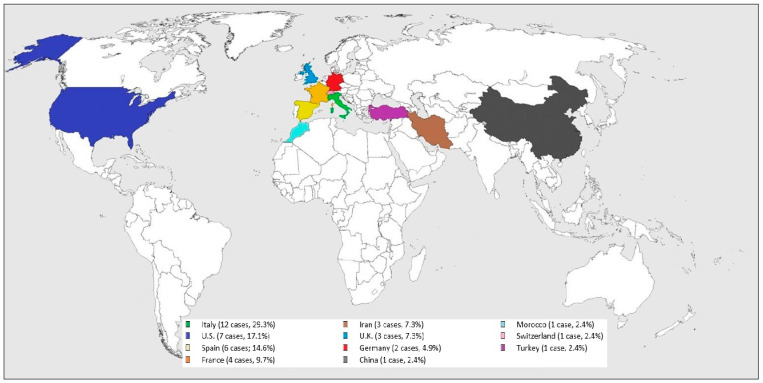
Worldwide SARS-CoV-2 related GBS distribution.

**Figure 3 life-11-00167-f003:**
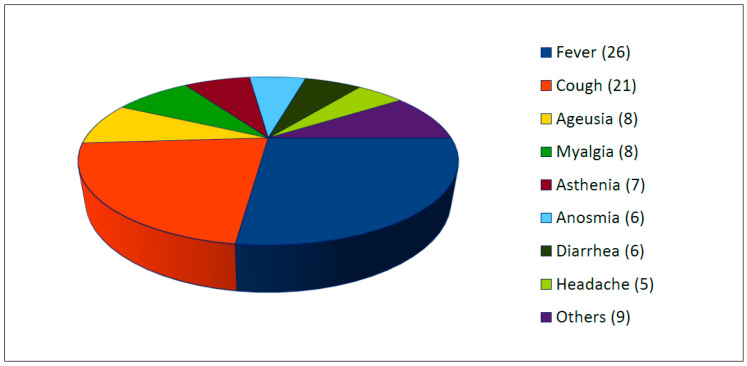
Clinical symptoms of SARS-CoV-2 infection (number, n =).

**Table 1 life-11-00167-t001:** Case reports of Guillain-Barre’ syndrome (GBS) with recent COVID-19 infection.

Authors, Year	Country	Sex, Age (Years)	COVID-19 Related Symptoms	COVID-19 Diagnosis	Neurological Symptoms	GBS Diagnosis	GBS Treatment	Outcome
***Alberti P. et al., 2020*** [10]	*Italy*	M, 71	Fever.	- **Nasal swab:** RT-PCR + - **CT scan:** ground-glass opacities.	Paresthesia at limb extremities and distal weakness. Flaccid severe tetraparesis.	- **CSF:** ACD. SARS-CoV-2 neg. - **Reflex:** Absent deep tendon reflexes. - **EMG:** AIDP.	IVIg (0.4 g/kg/d) for 5 days.	Died.
***Arnaud S. et al., 2020*** [11]	*France*	M, 64	Cough, fever, dyspnea and diarrhea.	- **Nasal swab:** RT-PCR + - **CT scan:** ground-glass opacities.	Distal weakness, flaccid paraparesis, decreased proprioceptive length-dependent sensitivity involving the four limbs.	- **CSF:** ACD. SARS-CoV-2 neg. - **Reflex:** Absent deep tendon reflexes. - **EMG:** AIDP.	IVIg (?) for 5 days.	Recovery.
***Assini A. et al., 2020*** [12]	*Italy*	M, 55	Anosmia, ageusia, fever and cough.	- **Nasal swab:** RT-PCR +	Bilateral eyelid ptosis, dysphagia, and dysphonia.	- **CSF:** Total protein level normal. SARS-CoV-2 neg. - **Reflex:** general hyporeflexia. - **EMG:** AIDP.	IVIg (0.4 g/kg/d) for 5 days.	Recovery.
		M, 60	Fever and cough.	**- Nasal swab:** RT-PCR + - **CT scan:** interstitial pneumonia.	Distal weakness, gastroplegia, paralytic ileus, and loss of blood pressure control.	- **CSF:** total protein level normal. SARS-CoV-2 neg. - **Reflex:** hyporeflexia. - **EMG:** AMSAN.	IVIg (0.4 g/kg/d) for 5 days.	Recovery.
***Bigaut K. et al., 2020*** [13]	*France*	M, 43	Cough, asthenia, myalgia, anosmia, ageusia, diarrhea.	- **Nasal swab:** RT-PCR + **- CT scan:** ground-glass opacities.	Paraesthesia, hypoesthesia, and distal weakness in the lower limbs. Ataxia. Right peripheral facial palsy.	- **CSF:** ACD. SARS-CoV-2 neg. - **Reflex:** Absent deep tendon reflexes (exept left biceps reflex). - **EMG:** AMAN.	IVIg (2 g/kg).	Recovery.
		F, 70	Anosmia, ageusia, diarrhea, mild asthenia.	- **Nasal swab:** RT-PCR + - **CT scan:** ground-glass opacities.	Flaccid tetraparesis, generalized areflexia, forelimb paresthesia, respiratory failure	- **CSF:** ACD. SARS-CoV-2 neg. - **Reflex:** Absent deep tendon reflexes. - **EMG:** AMSAN.	IVIg (2 g/kg).	Recovery.
***Caamaño DSJ et al., 2020*** [14]	*Spain*	M, 61	Cough, fever.	- **Nasal swab:** RT-PCR +	Peripheral facial nerve palsy.	- **CSF:** ACD. SARS-CoV-2 neg. - **Reflex:** Normal. - **EMG:** Not performed.	Prednisone.	Recovery.
***Camdessanche JP et al., 2020*** [15]	*France*	M, 64	Cough, fever.	- **Nasal swab:** RT-PCR + - **CT scan:** ground-glass opacities.	Paresthesia, flaccid severe tetraparesia. Swallowing disturbance.	- **CSF:** ACD. - **Reflex:** Absent deep tendon reflexes. - **EMG:** AIDP.	IVIg (0.4 g/kg/d) for 5 days.	?
***Coen M et al., 2020*** [16]	*Switzerland*	M, 70	Myalgia, fatigue, cough.	- **Nasal swab:** RT-PCR + - Anti-SARS-CoV-2 IgA and IgG +	Paraparesis, distal allodynia, difficulties in voiding, constipation.	- **CSF:** ACD. - **Reflex:** Absent deep tendon reflexes. - **EMG:** AIDP.	IVIg (0.4 g/kg/d) for 5 days.	Recovery.
***Dinkin M et al., 2020*** [17]	*U.S.*	M, 36	Fever, cough, and myalgias.	- **Nasal swab:** RT-PCR +	Left ptosis, diplopia, and bilateral distal leg paresthesias.	- **CSF:** Total protein level normal. SARS-CoV-2 neg. - **Reflex:** General hyporeflexia. - **EMG:** Not performed.	IVIg (2g/kg) for 3 days.	Recovery.
		F, 71	Cough and fever.	- **Nasal swab:** RT-PCR +	Diplopia.	- **CSF:** Normal. - **Reflex:** Not performed. - **EMG:** Not performed.	None.	Recovery.
***El Otmani H et al., 2020*** [18]	*Marocco*	F, 70	Cough.	- **Nasal swab:** RT-PCR + - **CT scan:** ground-glass opacities.	Weakness and tingling sensation in four extremities. Quadriplegia and hypotonia.	- **CSF:** ACD. SARS-CoV-2 neg. - **Reflex:** Absent deep tendon reflexes. - **EMG:** AMSAN	IVIg (2 g/kg) for 5 days.	No significant neurological improvement after one week.
***Fernandez-Dominguez J et al., 2020*** [19]	*Spain*	F, 74	Respiratory symptoms.	- **Nasal swab:** RT-PCR +	Gait impairment, blurred vision.	- **CSF:** ACD. - **Reflex:** Absent deep tendon reflexes. - **EMG:** ?	IVIg (20 g/d) for 5 days.	Recovery.
***Galan AV et al., 2020*** [20]	*Spain*	F, 43	Diarrhea and respiratory symptoms.	- **Nasal swab:** RT-PCR +	Distal weakness and gait impairment. Facial palsy and dysphagia.	- **CSF:** Not performed. - **Reflex:** General hyporeflexia. - **EMG:** AIDP	IVIg (?) for 5 days.	Recovery.
***Granger A et al., 2020*** [21]	*U.S.*	M, 48	Viral syndrome.	- **Nasal swab:** RT-PCR +	Progressive and symmetric sensorimotor deficits involving the face, extremities, and trunk. Right facial paralysis.	- **CSF:** ACD. - **Reflex:** Absent deep tendon reflexes. - **EMG:** AMSAN.	PLEX (5 sessions).	Recovery.
***Gutierrez-Ortiz C et al., 2020 *** [22]	*Spain*	M, 50	Cough, malaise, headache, low back pain, fever.	- **Nasal swab:** RT-PCR + - **Chest X-ray:** normal.	Vertical diplopia, perioral paresthesias, and gait instability.	- **CSF:** ACD. SARS-CoV-2 neg. - **Reflex:** Absent deep tendon reflexes. - **EMG:** Not performed.	IVIg (0.4 g/kg) for 5 days.	Recovery.
		M, 39	Diarrhea and fever.	- **Nasal swab:** RT-PCR + - **Chest X-ray:** normal.	Diplopia.	- **CSF:** ACD. SARS-CoV-2 neg. - **Reflex:** Absent deep tendon reflexes. - **EMG:** Not performed.	None.	Recovery.
***Lantos JE et al., 2020*** [23]	*U.S.*	M, 36	Fevers, chills, and myalgia.	- **Nasal swab:** RT-PCR +	Left eye drooping, blurry vision, reduced sensation and paresthesia in both legs. Ophthalmoparesis and ataxia.	- **CSF:** Not performed. - **Reflex:** general hyporeflexia. - **EMG:** Not performed.	IVIg (?)	Recovery.
***Manganotti P et al., 2020*** [24]	*Italy*	F, 50	Fever, cough and augesia.	- ?	Diplopia and facial paresthesia. Ataxia and gait impairment.	- **CSF:** ACD. - **Reflex:** ? **- EMG:** not performed.	IVIg (0.4 g/kg) for 5 days.	Recovery.
***Marta-Enguita J et al., 2020*** [25]	*Spain*	F. 74	Fever.	- **Nasal swab:** RT-PCR + - **CT scan:** interstitial pneumonia.	Progressive tetraparesis with distal paresthesias. Dysphagia.	- **CSF:** ACD. SARS-CoV-2 neg. - **Reflex:** Absent deep tendon reflexes. - **EMG:** Not performed.	?	Died.
***Naddaf E et al., 2020*** [26]	*U.S.*	F, 58	Dysgeusia, fever, myalgia, and asthenia.	- **Nasal swab:** RT-PCR - - Anti-SARS-CoV-2 IgA and IgG +	Progressive bilateral paraparesis, imbalance, severe lower thoracic pain and gait difficulty.	- **CSF:** ACD. SARS-CoV-2 neg. - **Reflex:** Absent in the legs and decreased in the upper extremities. - **EMG:** AIDP.	PLEX (5 sessions).	Slightly ataxic.
***Oguz-Akarsu E et al., 2020*** [27]	*Turkey*	F, 53	Fever.	- **Nasal swab:** RT-PCR + - **CT scan:** ground-glass opacities.	Dysarthria, progressive weakness and numbness of the lower extremities.	- **CSF:** ACD. SARS-CoV-2 neg. - **Reflex:** Absent in the legs. - **EMG:** AIDP.	PLEX (5 sessions).	Recovery.
***Ottaviani D et al., 2020*** [28]	*Italy*	F, 66	Fever and cough. Pruriginous dorsal rash.	- **Nasal swab:** RT-PCR – (I°), then + (II°). - **CT scan:** ground-glass opacities.	Difficulty walking and acute fatigue. Paraparetic with a rapidly progressive symmetric weakness in the upper and lower limbs.	- **CSF:** ACD. - **Reflex:** Absent deep tendon reflexes. - **EMG:** AIDP/AMAN	IVIg ( 0,4 g/kg) for 5 days.	Multiple Organ Failure.
***Padroni M et al., 2020*** [29]	*Italy*	F, 70	Fever and cough.	- **Nasal swab:** RT-PCR + (I°), then - (II°). - **CT scan:** ground-glass opacities.	Asthenia, hands and feet paresthesia and gait difficulties.	- **CSF:** ACD. SARS-CoV-2 neg. - **Reflex:** Absent deep tendon reflexes. - **EMG:** AIDP	IVIg (400 mg/d) for 5 days.	Respiratory failure.
***Paybast S et al., 2020*** [30]	*Iran*	M, 38	Viral syndrome	?	Ascending paresthesia, bilateral facial droop. Autonomic features (tachycardia and blood pressure instability). Swallowing disturbance.	- **CSF:** ACD. SARS-CoV-2 neg. - **Reflex:** Absent deep tendon reflexes. - **EMG:** AIDP/AMAN.	PLEX (5 sessions).	?
		F, 14	Headaches and dizziness.	?	Ascending quadripareshtesia. Lower limb weakness.	- **CSF:** ACD. SARS-CoV-2 neg. - **Reflex:** Hypoactive in upper limbs and absent in lower limbs. - **EMG:** Not preformed.	IVIg (20 g/d) for 5 days.	?
***Pfefferkorn T et al., 2020*** [31]	*Germany*	M, 51	Fever and flu-like symptoms with marked fatigue and cough.	- **Nasal swab:** RT-PCR + - **CT scan:** ground-glass opacities.	Progressive upper and lower limb weakness and acral paresthesias. Respiratory faiulure. Peripheral locked-in syndrome with tetraplegia.	- **CSF:** Total protein level normal. SARS-CoV-2 neg. - **Reflex:** General hyporeflexia. - **EMG:** AMSAN.	IVIg (30 g/d) for 5 days. PLEX (14 sessions).	Rehabilitation.
***Rana S et al., 2020*** [32]	*U.S.*	M, 54	Rhinorrhea, odynophagia, fevers, chills, and night sweats.	- **Nasal swab:** RT-PCR +	Ascending limb weakness and numbness. Difficulty voiding urine. Respiratory faiulure. Facial diplegia, quadriparesis and mild ophthalmoparesis.	- **CSF:** Not performed. - **Reflex:** Absent deep tendon reflexes. - **EMG:** AIDP/AMAN	IVIg (400 mg/kg/d).	Rehabilitation.
***Ray A, 2020*** [33]	*UK*	M, 63	Fever.	- **Nasal swab:** RT-PCR +	Diplopia, perioral paresthesias, finger tingling and gait impairment.	- **CSF:** ACD. - **Reflex:** Absent deep tendon reflexes. - **EMG:** Not performed.	None.	Recovery.
***Riva N et al., 2020*** [34]	*Italy*	M, 60	Fever, headache, myalgia, anosmia and ageusia.	- **Nasal swab:** RT-PCR - - Anti-SARS-CoV-2 IgG+ - **CT scan:** ground-glass opacities.	Progressive limb weakness and distal paresthesia at four-limbs. Facial diplegia, hypophonia and dysarthria.	- **CSF:** Normal. SARS-CoV-2 neg. - **Reflex:** Absent deep tendon reflexes. - **EMG:** AIDP.	IVIg (0.4 g/kg/d) for 5 days.	Recovery.
***Scheidl E et al., 2020*** [35]	*Germany*	F, 54	Anosmia and ageusia.	- **Nasal swab:** RT-PCR +	Proximally and symmetric paraparesis. Numbness and tingling of all extremities.	- **CSF:** ACD. - **Reflex:** Absent deep tendon reflexes. - **EMG:** AIDP.	IVIg (0.4 g/kg/d) for 5 days.	Recovery.
***Sedaghat Z et al., 2020*** [36]	*Iran*	M, 65	Cough, fever and dyspnea.	- **Nasal swab:** RT-PCR + - **CT scan:** ground-glass opacities.	Acute progressive symmetric ascending quadriparesis, facial paresis and dysphagia.	- **CSF:** Not performed. - **Reflex:** Absent deep tendon reflexes. - **EMG:** AMSAN.	IVIg (0.4 g/kg/d) for 5 days.	?
***Su XW et al., 2020*** [37]	*U.S.*	M, 72	Diarrhea, anorexia and chills.	- **Nasal swab:** RT-PCR +	Symmetric paresthesias and ascending appendicular weakness. Respiratory faiulure. Dysautonomia with hypotension alternating with hypertension and tachycardia. SIADH.	- **CSF:** ACD. SARS-CoV-2 neg. - **Reflex:** Absent deep tendon reflexes. - **EMG:** AIDP.	IVIg (2 g/kg) for 4 days.	Severe weakness.
***Tiet MY et al., 2020*** [38]	*U.K.*	M, 49	Shortness of breath, headache and cough.	- **Nasal swab:** RT-PCR +	Distal lower limb paraesthesia resulting in difficulty mobilising, facial diplegia, weakness and dysaesthesia in lower limbs.	- **CSF:** ACD. SARS-CoV-2 neg. - **Reflex:** Absent deep tendon reflexes. - **EMG:** AIDP.	IVIg (0.4 g/kg/d) for 5 days.	Recovery.
***Toscano G et al., 2020*** [39] *** **- Nasal swab:** *RT-PCR + (n = 4)* *- IgG + (n = 1)* ***** EMG:*** AIDP (n = 2); AMAN (n = 3)	*Italy*		Fever, cough, and ageusia	- **Nasal swab:** *RT-PCR + (n=4); - IgG + (n=1).*	Flaccid areflexic tetraplegia. Facial weakness, upper-limb paresthesia and respiratory faiure.	- **CSF:** I°) normal; II°) ACD. SARS-CoV-2 neg. - **EMG:** *	IVIg (?)	Severe upper-limb weakness, dysphagia, and lower-limb paraplegia.
			Fever and pharyngitis.		Facial diplegia and lower limb paresthesia with ataxia.	- **CSF:** ACD. SARS-CoV-2 neg. - **Reflex:** Absent deep tendon reflexes. - **EMG:** *	IVIg (?)	Recovery.
			Fever and cough.		Flaccid tetraparesis and facial weakness.	- **CSF:** ACD. SARS-CoV-2 neg. - **Reflex:** Absent deep tendon reflexes. - **EMG:** *	IVIg (?)	Respiratory failure and flaccid tetraplegia.
			Cough and hyposmia.		Flaccid tetraparesis and ataxia.	- **CSF:** normal. SARS-CoV-2 neg. - **Reflex:** Absent deep tendon reflexes. - **EMG:** *	IVIg (?)	Mild improvement.
			Cough, ageusia, and anosmia.		Facial weakness and flaccid paraplegia.	- **CSF:** ACD. SARS-CoV-2 neg. - **Reflex:** Absent deep tendon reflexes. - **EMG:** *	IVIg (?) PLEX (?)	Bacterial pneumonia.
	**** EMG:*** AIDP (n = 2); AMAN (n = 3)							
***Webb S et al., 2020*** [40]	*U.K.*	M, 57	Cough, headache, myalgia and malaise.	- **Nasal swab:** RT-PCR + - **CT scan:** ground-glass opacities.	Progressive limb weakness and foot dysaesthesia. Respiratory faiulure.	- **CSF:** ACD. SARS-CoV-2 neg. - **Reflex:** Hyporeflexia. - **EMG:** AIDP.	IVIg (2 g/kg/d) for 5 days.	Mild improvement.
***Zhao H et al., 2020*** [41]	*China*	F, 61	Cough and fever.	- **Nasal swab:** RT-PCR + - **CT scan:** ground-glass opacities.	Acute weakness in both legs and severe fatigue.	- **CSF:** ACD. SARS-CoV-2 neg. - **Reflex:** Absent deep tendon reflexes. - **EMG:** AIDP	IVIg (?)	Recovery.

ACD, albuminocytological dissociation; AIDP, acute inflammatory demyelinating polyradiculoneuropathy; AMAN, acute motor axonal neuropathy; AMSAN, acute motor and sensory axonal neuropathy; CSF, cerebrospinal fluid; CT, computed tomography; EMG, electromyography; F, female; IVIg, intravenous immunoglobulin; M, male; PCR, polymerase chain reaction; PLEX, plasma exchange; SARS-CoV-2, severe acute respiratory syndrome coronavirus-2; UK, United Kingdom; US, United States; ?, not reported.

**Table 2 life-11-00167-t002:** Neurological symptoms and signs.

**Neurological Symptoms**	**N**	**N/GBS Cases * (%)**
Dysphagia	7	17.1
Dysarthria	4	9.8
Diplopia	9	21.9
**Neurological Signs**	**N**	**N/GBS Cases * (%)**
Facial palsy	13	31.7
Bulbar palsy	4	9.8
Ocular palsy	7	17.1
Tetraparesis	11	26.8
Paraparesis	7	17.1
Paresthesia	26	63.4
Areflexia or hyporeflexia	34	82.9
Ataxia	12	29.3
Respiratory dysfunction	7	17.1
Dysautonomia	5	12.2

* GBS cases = 41.

**Table 3 life-11-00167-t003:** Ancillary investigations, treatment, and disease progression of GBS cases associated with SARS-CoV-2.

**SARS-CoV-2 virus Certainty Level**	**Cases (%)**		
Confirmed	40/41 (97.6%)		
Probable	1/41 (2.4%)		
Suspected	0/41		
**Arboviral Tests**	**Cases (%)**		
SARS-CoV-2 virus	Total	Pos.	Neg.
PCR (Nasopharyngeal swab test)	36/41 (87.8%)	33/36 (91.7%)	3/36 (8.3%)
PCR (Oropharyngeal swab test)	4/41 (9.7%)	4/4 (100%)	0/4
IgA serum	2/41 (4.9%)	2/2 (100%)	0/2
IgG serum	4/41 (9.7%)	4/4 (100%)	0/4
PCR CSF	24/41 (58.5%)	0/24	24/24 (100%)
Serological test (Campylobacter jejuni, HIV, syphilis, CMV, and EBV)	11/41 (26.8%)	0/11	11/11 (100%)
**Radiology Test**	**Cases (%)**		
COVID-19 radiologiac features	Total	Pos.	Neg.
Chest X-ray	11/40 (27.5%)	5/11 (45.5%)	6/11 (54.5%)
Chest CT	15/40 (37.5%)	15/15 (100%)	0/15
**CSF Analysis**			
Increased protein level	1/31 (3.2%)		
ACD	26/31 (83.9%)		
Normal	4/31 (12.9%)		
**Electrophysiological Exam**			
AIDP	17/30 (56.7%)		
AMAN	4/30 (13.3%)		
AMSAN	5/30 (16.7%)		
Equivocal	3/30 (10.0%)		
Inconclusive	1/30 (3.3%)		
**Immunomodulatory Treatment**			
IVIg	30/36 (83.3%)		
Plasma exchange	4/36 (11.1%)		
IVIg and plasma exchange	2/36 (5.6%)		
**Disease Progression**			
Admission to ICU	14/41 (34.1%)		
Mechanical ventilation	14/41 (34.1%)		
Died	2/41 (4.9%)		

ACD, albuminocytological dissociation; AIDP, acute inflammatory demyelinating polyradiculoneuropathy; AMAN, acute motor axonal neuropathy; AMSAN, acute motor and sensory axonal neuropathy; CMV, cytomegalovirus; CSF, cerebrospinal fluid; CT, computed tomography; EBV, Epstein-Barr virus; HIV, human immunodeficiency virus; ICU, intensive care unit; IVIg, intravenous immunoglobulin; PCR, polymerase chain reaction.

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
