# Peer review of "Post-Infectious Guillain–Barré Syndrome Related to SARS-CoV-2 Infection: A Systematic Review"

_life, 2021, doi:10.3390/life11020167_

Round 1

Reviewer 1 Report

Sansone et al reported a review of 41 patients with post-infectious Guillain–Barré syndrome related to SARS-CoV-2 infection published since the onset of the pandemic worldwild. This is an interesting work highlighting the Relationship between viruses and Guillain-Barré syndrome.

However, I have some comments. 

Minor revisions :

Abstract :

-line 24 : please provide a number or percentage of cases of patients who responded to Ig IV

-line 26 : rather than « rarely » provide a number or a percentage.

Introduction :

- line 64 : authors probably meant « systematic » rather than « systemic »

Results :

-line 143 : authors wrote « GBS was diagnosed in 36 cases » : this is not clear since there are 41 patients with GBS... what about the 5 missing cases ?

-line 171, caption of the Figure 3 : what is the « (n) » ?

-line 177 : authors wrote « neurologic » and should change this by "neurological": please homogeneize in all the manuscript (and table)

Major revisions :

Introduction :

Line 38-39 : « Recently, the incidence of GBS has increased during the severe acute respiratory syn-38 drome coronavirus-2 (SARS-CoV-2) epidemics worldwide » : please provide a reference for this assertion

Results :

-Figure 1 is missing as well as all the figures...

-In the « Study characteristics » part, line 127-130 : the total is 40 not 41

-Table 2 : please precise the number of GBS because values are not acute if authors considered that there were 41 patients with GBS

-Table 3 : one patient is missing in the « SARS-CoV-2 virus certainty level » part ; authors wrote that there were 40 patients in total, why ?

Discussion :

-Line 242 : Was there intrathecal synthesis of Ig in the CSF ? authors should add this information in the Results section and discuss it in the discussion.

-line 257-258 : is there an explanation or hypothesis about the predominance of elderly cases ?

Conclusions : line 300 : Authors can not conclude this point about prevalence since this is a retrospective study...

Author Response

Dear reviewer,
thanks for the review. Below are the required changes.

Abstract :

-line 24 : please provide a number or percentage of cases of patients who responded to Ig IV → “In the 75% the cases the response to immunomodulating therapy is favourable.”

-line 26 : rather than « rarely » provide a number or a percentage. → Rarely the outcome is poor or even fatal (10.8% of the cases).

Introduction :

- line 64 : authors probably meant « systematic » rather than « systemic » → Modified

Results :

-line 143 : authors wrote « GBS was diagnosed in 36 cases » : this is not clear since there are 41 patients with GBS... what about the 5 missing cases ? → Modified.

-line 171, caption of the Figure 3 : what is the « (n) » ? → (number, n =).

-line 177 : authors wrote « neurologic » and should change this by "neurological": please homogeneize in all the manuscript (and table) → Modified.

Introduction :

Line 38-39 : « Recently, the incidence of GBS has increased during the severe acute respiratory syn-38 drome coronavirus-2 (SARS-CoV-2) epidemics worldwide » : please provide a reference for this assertion → “Recently, several cases of GBS were reported during the severe acute respiratory syndrome coronavirus-2 (SARS-CoV-2) epidemics worldwide.”

Results :

-Figure 1 is missing as well as all the figures... → They are reported in a separated file.

-In the « Study characteristics » part, line 127-130 : the total is 40 not 41 → Modified.

-Table 2 : please precise the number of GBS because values are not acute if authors considered that there were 41 patients with GBS → Done.

-Table 3 : one patient is missing in the « SARS-CoV-2 virus certainty level » part ; authors wrote that there were 40 patients in total, why ? → Modified.

Discussion :

-Line 242 : Was there intrathecal synthesis of Ig in the CSF ? authors should add this information in the Results section and discuss it in the discussion. → We reported: “None of the patients with reported CSF analysis had SARS-CoV-2 in the CSF. This is in contradiction with various case reports, which detected SARS-CoV-2 in the CSF [49,50]. A possible explanation is the lack of disruption of the blood-brain barrier that would allow SARS-CoV-2 to cross the CSF space [51].”

-line 257-258 : is there an explanation or hypothesis about the predominance of elderly cases ? → As with non‐SARS‐CoV‐2‐associated GBS, elderly patients are more frequently affected than the younger.

Conclusions : line 300 : Authors can not conclude this point about prevalence since this is a retrospective study... → Modified: “It is important to be aware of this association to avoid delay in diagnosis and to promote early treatment initiation given the significant risk of morbidity and mortality.”

Reviewer 2 Report

Sansone et al performed a systematic review to identify published GBS cases in patients with SARS-CoV-2 infection. I read this work with great interest; this is a relevant topic, as we are still going through the pandemic worldwide. Their conclusion is similar to other reviews on this topic: GBS due to SARS-CoV-2 infection seems to share several clinical, laboratoristics and neurophysiological features with GBS from other infections. 

Overall, the manuscript is well written and clear. It could be further improved with some changes, though.

Title: I'd suggest "Guillain-Barré syndrome related to SARS-CoV-2 infection" would be a better option because, as you state in the text, the exact mechanism of roots and nerve involvement (para-infectious? post-infectious?) is still unknown.

Abstract is ok.

Introduction: "... this is the first published systemic review of GBS secondary to COVID-19." I think you should re word this, as I found already published systematic reviews about this topic. Highlight what your review brings to the current literature. 

Methods

  • Include PRISMA check-list in supplementary material;
  • "Predefined exclusion criteria were: GBS within 3 months after a vaccination or other proven triggering infection (e.g. Campylobacter jejuni), and studies with no information on age, residence, and at least one clinical variable of interest"; article by Toscano et al was included, even though it did not report patients age. Please clarify this point.

Results

  • text and figure 2 are incorrect; cases from Italy were 12, not 11. Please modify accordingly;
  • there are some discrepancies between text and tables (please clarify):
    • oropharyngeal swab tests: three positive in text (page 4), four positive in table 3;
    • probable SARS-CoV-2 infection: one in text (page 5), two in table 3
    • CSF examined in 21 patients according to the text (page 6), 31 according to table 3
  • Ageusia correct percentage is 22.0% in "clinical characteristics". 
  • Regarding table 2 and text, I would argue that "dysarthria" is not a symptom but a sign, and "paresthesia" is a symptom. Also, table percentages are all incorrect. Please modify
  • For table 3, write a legend with all acronyms explained. 
  • Table 3: can mean/range values be provided for abnormal protein and cell counts? This would be helpful and interesting.

Discussion:

  • Re word the first sentence: "It is now known that SARS-CoV-2 can often affect central and peripheral nervous system, apart from the bronchopulmonary system."
  • "The current study demonstrates a significantly higher frequency of SARS-CoV-2 infection in GBS patients." The statement is incorrect: this study does not provide epidemiological features of SARS-CoV-2 infection in GBS patients. 

Author Response

Dear reviewer,
thanks for the review. Below are the required changes.

Introduction: "... this is the first published systemic review of GBS secondary to COVID-19." I think you should re word this, as I found already published systematic reviews about this topic. Highlight what your review brings to the current literature. → “To help add to this small albeit developing body of evidence, this systematic review adds to other studies on GBS secondary to COVID-19.”

Methods

  • Include PRISMA check-list in supplementary material; → Included.
  • "Predefined exclusion criteria were: GBS within 3 months after a vaccination or other proven triggering infection (e.g. Campylobacter jejuni), and studies with no information on age, residence, and at least one clinical variable of interest"; article by Toscano et al was included, even though it did not report patients age. Please clarify this point. → “Predefined exclusion criteria were: GBS within 3 months after a vaccination or other proven triggering infection (e.g., C. jejuni), and studies with no information on residence, and at least one clinical variable of interest.”

Results

  • text and figure 2 are incorrect; cases from Italy were 12, not 11. Please modify accordingly; → Corrected.
  • there are some discrepancies between text and tables (please clarify):
    • oropharyngeal swab tests: three positive in text (page 4), four positive in table 3; → “four for oropharyngeal swab test for SARS-CoV-2 by qualitative RT-PCR assay”
    • probable SARS-CoV-2 infection: one in text (page 5), two in table 3 → table modified.
    • CSF examined in 21 patients according to the text (page 6), 31 according to table 3 à “CSF was examined in thirty-one studies”
  • Ageusia correct percentage is 22.0% in "clinical characteristics". → “ageusia (22%, 9/41)”
  • Regarding table 2 and text, I would argue that "dysarthria" is not a symptom but a sign, and "paresthesia" is a symptom. Also, table percentages are all incorrect. Please modify → Modified.
  • For table 3, write a legend with all acronyms explained. → Added.
  • Table 3: can mean/range values be provided for abnormal protein and cell counts? This would be helpful and interesting.

Discussion:

  • Re word the first sentence: "It is now known that SARS-CoV-2 can often affect central and peripheral nervous system, apart from the bronchopulmonary system." → Done.
  • "The current study demonstrates a significantly higher frequency of SARS-CoV-2 infection in GBS patients." The statement is incorrect: this study does not provide epidemiological features of SARS-CoV-2 infection in GBS patients. → “The current study reports various cases of GBS or its variants in patients with SARS-CoV2 infection.”

Round 2

Reviewer 1 Report

Authors correctly responded to my comments and performed appropriate changes.